# Words, Subwords, and Morphemes: What Really Matters in the Surprisal-Reading Time Relationship?

**Sathvik Nair** and **Philip Resnik**
Linguistics & Institute for Advanced Computer Studies
University of Maryland
{sathvik,resnik}@umd.edu

## Abstract

An important assumption that comes with using LLMs on psycholinguistic data has gone unverified. LLM-based predictions are based on subword tokenization, not decomposition of words into morphemes. Does that matter? We carefully test this by comparing surprisal estimates using orthographic, morphological, and BPE tokenization against reading time data. Our results replicate previous findings and provide evidence that *in the aggregate*, predictions using BPE tokenization do not suffer relative to morphological and orthographic segmentation. However, a finer-grained analysis points to potential issues with relying on BPE-based tokenization, as well as providing promising results involving morphologically-aware surprisal estimates and suggesting a new method for evaluating morphological prediction.

## 1 Introduction

There is widespread consensus that human sentence processing includes word-level prediction (Ehrlich and Rayner, 1981); see Staub (2015) for a review. A growing body of research is making use of language models as computational proxies for human prediction at the word level, including traditional $n$-gram models (Smith and Levy, 2013), syntax-based models (Hale, 2001; Demberg and Keller, 2008), and more recent work that makes use of neural language models (Goodkind and Bicknell, 2018; Wilcox et al., 2020; Shain et al., 2022).

As an overall paradigm, research in this area generally correlates *surprisal*, computed using corpus-based probability estimates ($-log \Pr(\text{word}|\text{context})$), against measurable indices of human processing effort. These include measurement of processing activity using fMRI (Henderson et al., 2016; Bhattasali and Resnik, 2021; Shain et al., 2020), MEG (Donhauser and Baillet, 2020; Brodbeck et al., 2022), EEG (Rabovsky et al., 2018; Li and Ettinger, 2023;

Michaelov et al., 2023), and reading times (Demberg and Keller, 2008; Smith and Levy, 2013; Goodkind and Bicknell, 2018; Wilcox et al., 2020; Shain et al., 2022). This correlation paradigm has produced useful insights about the role of prediction in language comprehension. In addition, correlations between language model surprisal and human processing activity have been taken to be an indication that language models are capturing relevant aspects of how human language works (Ryu and Lewis, 2021; Goldstein et al., 2022).[1]

Lost in the shuffle of this research progress, however, is the question of what, exactly, "prediction at the word level" is supposed to mean. Within current linguistic theory, the very construct of "word" is receiving critical attention: albeit controversially, convincing arguments exist that the term *word* lacks a consistent, scientifically valid definition that holds consistently across the full range of human languages (Halle and Marantz, 1993; Haspelmath, 2017); see Krauska and Lau (2023) for an overview connecting these theoretical claims to psycholinguistics and neurolinguistics. Even setting aside that theoretical debate, however, the move to psycholinguistics and neurolinguistics using large language models has generated a mismatch between the units analyzed in human studies — typically orthographic words — and the subword tokens over which LLMs operate via approaches like Word-Piece and Byte-Pair Encoding (Wu et al., 2016; Sennrich et al., 2016).

To take an example, a typical LLM tokenization of the word *decomposition* using a widely used BPE tokenization (Sennrich et al., 2016) yields subword units dec, om, position.[2] In a typical

---

[1] These two kinds of results are important to distinguish. In the first case, human beings and their cognitive systems are the object of study. The second is an instance of what Simon (1969) referred to as "science of the artificial", where in this case the constructed computational system is itself the object of study.

[2] We use the GPT2 implementation of BPE throughout.

human subjects experiment, measurements such as first-pass reading time would typically involve a region from the beginning of the whole word to its end. More generally, the measurement of activity for a word $w$ in the human experimentation is related to language model predictions using $w$'s subword units $s_1 s_2 \ldots s_n$. How is a correlation computed between measurements over different units?

The solution adopted by most researchers (Hao et al., 2020; Wilcox et al., 2020; Stanojević et al., 2022, and others) is to compute surprisal separately for the $s_i$, and then approximate the model's surprisal for $w$ as the *sum* of those individual subword surprisals. The logic behind this choice is that, if the linking hypothesis behind the work connects surprisal with cognitive effort, the effort for the entire word should be the sum of the effort on each of the parts (Smith and Levy, 2013).[3]

This, however, leads to another question: is that a reasonable approximation, given that the subwords produced by LLM tokenization bear no clear correspondence to the subword decompositions in human processing? Consider again the word *decomposition*. A large body of theoretical and empirical work would suggest that to the extent subword effort takes place, it would involve *morphological* units, in this case de, compos(e), and (i)tion (Gwilliams, 2020).[4]

The question we set out to answer in this paper, then, is whether the divergence between LLM subword tokenization and human morphological decomposition is something to worry about in computational psycholinguistics and computational neurolinguistics research. Operationalizing the question, would the sum-of-surprisals approach with morphologically valid units yield a better correspondence with human measurements than the standard approach using LLM subword tokens? We would argue that the result is important regardless of which way the experimentation goes. If morphological units turn out to be a significantly better fit for human measurements, then cognitive researchers using LLMs should be using them — which could potentially raise real challenges given the widespread use of off-the-shelf pretrained

LLMs. If statistically-driven subword units work just as well, then we have checked an important, previously unchecked box in terms of validating their use.

This work is very much in the spirit of Wilcox et al. (2020), who evaluated a "fleet" of language models across architectures, plus orthographic $n$-gram models, against eyetracking and self-paced reading data. However to our knowledge, this study is the first to consider the assumption that LLM-subwords can be used in lieu of morphological units.

## 2 Methods

We trained three $n$-gram models under different word segmentation methods and evaluated them against reading time data from psycholinguistic experiments conducted in English. Our choice of evaluation corpora and metrics are consistent with previous literature, such that we were only examining the effect of word segmentation without model architecture as a confound. Our implementation, along with instructions for accessing the associated data, is available at https://github.com/sathvikn/dl-psych-tokenization/.

### 2.1 Language Models

Each $n$-gram model was a 5-gram model trained on the publicly available section of the Corpus of Contemporary American English (COCA, Davies (2010)). The models were trained under KenLM (Heafield, 2011) using modified Kneser-Ney smoothing (Ney et al., 1994). As a control, we used a model trained on COCA which was trained to predict the next orthographic word without any subword tokenization. To test BPE-based tokenization, we used the Huggingface implementation of the GPT2 tokenizer (Wolf et al., 2020; Radford et al., 2018) for each sentence in the corpus, and trained the $n$-gram model on individual tokens rather than words. Most current psycholinguistic work uses off-the-shelf GPT2 implementations and GPT2 (and variants) have been shown to be better fits to reading time data than larger models (Shain et al., 2022; Oh and Schuler, 2023). Finally, for a more linguistically informed approach to word segmentation, we trained an $n$-gram model based on the output of a morphological transducer (Wehrli et al., 2022) that is far more accurate than WordPiece tokenization at word and sentence-level morpheme segmentation tasks in a variety of lan-

---

[3]In principle one could be more sophisticated by calculating the uniqueness point within the word in a given lexicon (Luce, 1986) and only summing up assigned complexity values for wordpieces that precede that point.

[4]We discuss another illustrative example in Appendix A and Section 2.1 provides details on the morphological segmenter we use in our experimentation.

guages, including English (Batsuren et al., 2022).[5] The morphological transducer was based on an encoder-decoder architecture, which used a two-layer stacked LSTM as the encoder and performed greedy decoding. We judged the corpus too small for retraining a GPT-style model, and we did not train an LSTM because Wilcox et al. (2020) conclusively showed 5-gram models are stronger predictors of results in broad-coverage psycholinguistic experiments.[6]

## 2.2 Psycholinguistic Evaluation

Once we trained the models, we computed their surprisal estimates for words in eyetracking and self-paced reading corpora and fit regression models evaluating surprisal as a predictor of the reading times from these corpora.

### 2.2.1 Corpora

We used averaged eye movement data from the Dundee corpus (Kennedy et al., 2003) and self-paced reading times from the Natural Stories corpus (Futrell et al., 2018) made available by Wilcox et al. (2020). Both corpora are representative of the material in COCA. In the Dundee corpus, each word's fixation time in milliseconds was averaged across 10 English-speaking participants reading newspaper articles. The Natural Stories corpus consists of sentences from narrative texts that were edited to include syntactic constructions that are rare in spoken and written English for psycholinguistic analysis. The self-paced reading times were recorded from 181 English speakers who read the texts as they were presented word-by-word. We used the per-word presentation times that were averaged across participants.

### 2.2.2 Measuring Predictive Power of Surprisal

To compare the reaction times to the models, we computed the surprisal for each word under each $n$-gram model. The model trained on orthographic words generated a surprisal for each word, but since the BPE tokenizer and the morphological transducer used subword information, we tokenized the

texts from the behavioral experiments and computed each token's surprisal. If a word was split into multiple tokens, its surprisal under the other two models was the sum of the tokens' individual surprisals.[7] We then fit regression models predicting reading time based on surprisal.

For each word segmentation method, we compared the per-token log likelihood ($\Delta LogLik$) under two multiple linear regression models, following previous literature to quantify how much surprisal contributes to reading time prediction, independent of other predictors. One model used the control features of word length and log unigram frequency to predict reading times as a baseline model, and the other used these factors in conjunction with surprisal. If the regression model with surprisal-based features generated more accurate predictions, the difference between the log likelihoods would be above zero.

Although $\Delta LogLik$ as the measure of predictive power is standard for this literature, we highlight two specific methodological details from (Wilcox et al., 2020). First, the predictive power normalizes the regression models' aggregate log likelihood since we are comparing this metric on corpora with different sizes. Second, we used 10-fold cross validation to report $\Delta LogLik$ on a held-out test set. The training and testing data were consistent across all models for each fold. Reporting the value of a cross-validated regression model is important to ensure that the predictive power measures computed on the complete dataset are not the result of overfitting (Yarkoni and Westfall, 2017). Due to spillover effects from previous words (Smith and Levy, 2013), we also included the surprisal, length, and log frequency of the previous word as predictors in the the regression models for the Dundee corpus, and similarly for the previous three words for the Natural Stories corpus.[8]

## 3 Results

All our results show surprisal improves predictions of reading time relative to the control features, com-

---

[5]This analyzer came in second in the SIGMORPHON competition. We chose it because the first place system's implementation was not publicly available.

[6]In this study we used a publicly available subset of the COCA corpus for replicability, and it is less than 2/3 the size of the training dataset for the smallest orthographic GPT2 model in Wilcox et al. (2020). A near-term aim for future work is to replicate this experimentation with the full COCA corpus, which requires a license.

[7]We excluded data for certain words from our analysis following Goodkind and Bicknell (2018). These were words preceding and following punctuation, words that contained non-alphabetical characters, and words that were out of vocabulary for the language models. If any token under a language model was not in its vocabulary, the entire word was excluded.

[8]This difference arises because the corpora were used with different psycholinguistic tasks (Wilcox et al., 2020); we report results for each corpus separately for this reason, and because the corpora have different sizes and material.

| Tokenization Method | $\Delta LogLik$ for Dundee | $\Delta LogLik$ for Natural Stories |
|---|---|---|
| Orthographic | 0.01 | 0.009 |
| BPE | 0.01 | 0.01 |
| Morphological | 0.01 | 0.008 |

Table 1: Per-token $\Delta LogLik$ showing surprisal-based regression models improve over controls.

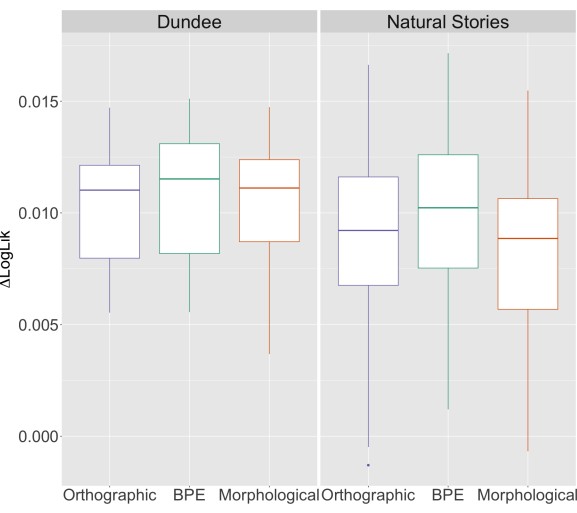

Figure 1: Distribution of predictive power of surprisal under models trained under each tokenization scheme, evaluated using 10-fold cross-validation for each corpus. There is no major difference in predictive power associated with tokenization.

parably to the 5-gram models' results in Goodkind and Bicknell (2018) and Wilcox et al. (2020). Table 1 reports the difference in per-token log likelihood between the linear regression models predicting words' reading times using surprisal as a feature and the models which simply used length and log frequency as features. We also report a more conventional measure of effect size using Cohen's $f^2$ in Table 5. The surprisals of the previous and current word were statistically significant predictors of reading time for all models on all corpora $(p < 0.001)$.[9]

We also report $\Delta LogLik$ on a held-out test set using 10-fold cross validation (Figure 1). The training and testing data were consistent across all models for each fold. Reporting the value of a cross-validated regression model is important to ensure that the predictive power measures computed on the complete dataset are not the result of overfitting. We compared the distribution of these values under a Wilcoxon rank-sum test (Table 6). The predictive power of surprisal under the models using BPE and the morphological transducer's output did not show a statistically significant difference from the model using orthographic words. For the Natural Stories corpus, the predictive power of surprisal was lower than the Dundee corpus, which is expected since the Natural Stories corpus contained rare syntactic constructions.

On the face of it, these results seem to show that LLM-style tokenization may not be an issue in psycholinguistic modeling. However, finer-grained analyses suggest otherwise. First, few words in the psycholinguistic corpora were split by the BPE tokenizer in the first place. As it turns out, the BPE tokenizer only split 5% of the tokens in the psycholinguistic corpora (11% when ignoring stopwords), compared to 25% and 44% respectively for the morphological analyzer (complete results

in Table 3). Moreover, the standard linking theory for surprisal suggests that effort for the entire word should be the sum of the subword efforts (Smith and Levy, 2013), and therefore that processing effort should increase incrementally with the number of units a word is segmented into. But this appears to be true only for the morphological tokenization: for BPE tokenization there is a sharp jump from low surprisal with unsplit words to essentially equal surprisal for words split into 2, 3, and 4 tokens (Figure 2). The data therefore suggest that surprisal based on BPE tokenization is less cognitively realistic than surprisal over morphological units. In addition, replicating the entire analysis separately for non-segmented and segmented words, we find that the predictive power of the BPE-based model is significantly worse for words that do get split by the tokenizer, and this is not true for the morpheme-based model (Figure 3). We conclude that, despite the aggregate results in Figure 1, LLM-style tokenization should be viewed with caution in cognitive investigations.

## 4 Conclusions

This study was a small, focused contribution that tackled an essential question previously unaddressed in psycholinguistics research that uses LLMs and their subword tokenizations. Previous work has demonstrated a linear relationship between LLM surprisal and human processing as measured by reading time, but there is good evidence

---

[9]For Natural Stories, the tokens at $w_{i-3}$ and $w_{i-2}$ also had some predictive power ($p < 0.01$ and $p < 0.1$, respectively).

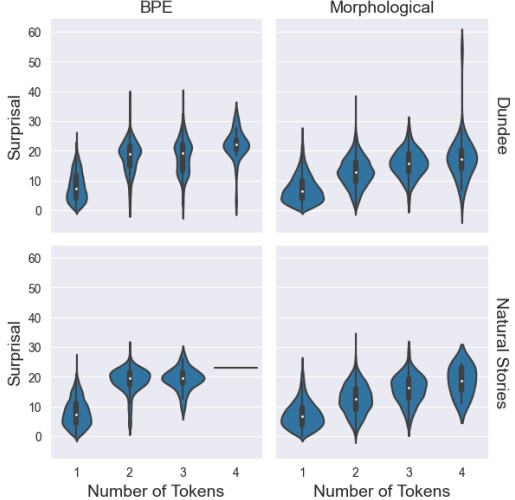

Figure 2: Distributions of surprisal of words with different numbers of subword tokens, split by corpus and segmentation method.

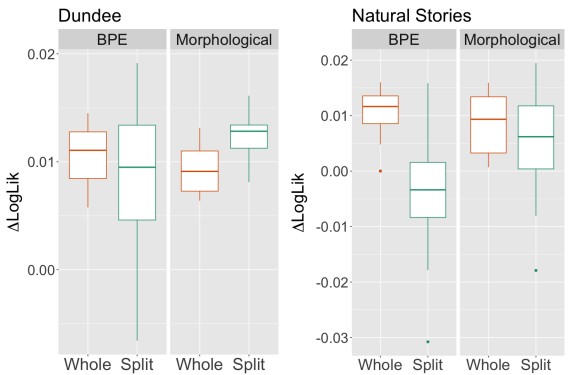

Figure 3: Replicating the cross-validated analysis of predictive power separately for whole and split words under the different word segmentation methods.

that aspects of human processing are mediated by *morphological* units. For cognitive research, mightn't it be important to model surprisal using morphological units, rather than distributionally derived subword tokens that often deviate drastically from a word's morphological decomposition?

On the one hand, our work offers the first comparison demonstrating that using distributionally derived rather than morphological subwords does not affect *aggregate* correlations between surprisal and reading times, a widely used behavioral measurement of human processing. On the other hand, given that psycholinguistics work is increasingly using proprietary pre-trained models with non-morphological subword tokenization, far beyond the scale available for academic model training, our finer-grained analysis indicates that a degree of caution is warranted for more fine-grained stud-

ies. Perhaps our most important take-away is that cognitive investigations require a careful look at the cognitive plausibility of the models' units of analysis.

Our results also suggest new directions for more cognitively realistic models of prediction in language comprehension. We view the results from our finer-grained analysis as a step in this direction, and they also suggest going more deeply into the role of morphological segmentation on an item-by-item basis; for example, training an LLM on morphological units and evaluating it on diagnostics for morphosyntactic generalization. Our work here also introduces *morphological surprisal* computed automatically using a morphological segmenter, and validates its predictive power. This would be a natural fit for further work on morphological prediction at the neural level (Gwilliams, 2020), including looking at the role of morphemes in phoneme prediction (Ettinger et al., 2014), and as a new representational level within integrative processing models that take phonological, word-level, and sentence-level contexts into account (Brodbeck et al., 2022). This implementation could be refined through inferring hierarchical structure over morphological units in the style of Oseki and Marantz (2020) to conduct larger-scale analyses.

Finally, regarding broader theoretical discussion, we note that surprisal (as operationalized using an LLM or any other probability estimates) generally contributes to explanations at Marr's "computational" level (Marr, 1982).[10] Moreover, LLM predictions represent a black-box combination of categories of information that both theoretical and experimental considerations suggest are processed in distinct ways (Oseki and Marantz, 2020; Oh et al., 2021; Krauska and Lau, 2023). We would therefore argue that, despite their undeniable convenience and power, the widespread use of LLMs as probabilistic predictors deserves drastically more careful consideration than it has received if the field is to move in the direction of deeper insights into human representations and mechanisms in language processing.

## 5 Limitations

The study was limited to English, and it is possible different results might obtain in languages

---

[10]As an interesting and promising exception, Futrell et al. (2020) take a step in the direction of the algorithmic/representational level by bringing memory considerations into the model.

with other morphological structures. However, the morphological transducer can be trained on any language for which a suitable morphologically segmented corpus is available, and it has already been evaluated on a multilingual test suite (Batsuren et al., 2022), so this is a promising topic for future work. This is an active area of research, especially because not all languages have the same notion of what counts as a "word." Existing work (de Varda and Marelli, 2022; Wilcox et al., 2023) has evaluated predictions on a multilingual eyetracking corpus with some typological diversity, but still trains transformer language models on subword tokenization. More work is also needed to see if results vary across mono- and multi-morphemic words; see Appendix B for an indication that LLM subword tokenizations can still be problematic at the level of individual predictions, even for words that do not include much morphological complexity. We also note that the publicly available version of COCA we used was preprocessed by Yang et al. (2022), and this may have led to some small discrepancies with the results from previous studies trained on other academically feasible datasets.

Perhaps our most significant limitation was in using $n$-gram versus state-of-the-art LLM architectures for our comparison, which in principle may not generalize to the best models. We would strongly encourage those who are able to train LLMs at scale to consider offering models with morphologically valid segmentation, both to facilitate more extensive language model comparisons, and to support scientific studies involving morphological representations as articulated in Section 4.

## 6   Ethical Considerations

All data we used are publicly available. Human experimentation was approved by the institutions who conducted the research, including our own. The software we used was publicly available, and the trained morphological segmenter was distributed by the authors of the paper, so our implementation and data analyses do not require specialized computing hardware.

## 7   Acknowledgements

This material is based upon work supported by ONR MURI Award N00014–18–1–2670. We would like to thank Silvan Wehrli for providing the English morphological segmentation model. John Hale, Tal Linzen, Brian Dillon, and Allyson Ettinger provided invaluable discussion as we formulated our research question, and Marine Carpuat, Utku Turk, and other members of the Computational Linguistics & Information Processing and Psycholinguistics groups at UMD provided insightful feedback on this work during various stages. Last, we thank the reviewers for suggesting improvements and clarifications to the paper, particulary comments and questions that motivated our finer-grained analysis.

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

## A    Comparing Word Segmentation Methods

In this illustrative example, the BPE-based tokenizer fails to split up most multimorphemic words. When it succeeds, the words are not segmented by morpheme (*relegated, fringes*). The morphological transducer is able to make cognitively plausible choices involving tense (*relegates*), possessives (*its*), and plurals (*fringes*). It also splits up more complex words into roots, prefixes, and suffixes (*coverage, reporters, journalistic, community*). Both tokenizers marked word boundaries in their output, although they are not shown in this example.

In Tables 3 and 4 we report the number of words that were split by both the BPE tokenizer and the morphological segmenter in the psycholinguistic corpora.

| Segmentation Method | Sentence |
|---|---|
| Orthographic Words | the sporadic nature of press coverage of the court often relegates its reporters to the fringes of the journalistic community |
| BPE Tokenizer Output | the sporadic nature of press coverage of the court often **releg ates** its reporters to the **fr inges** of the journalistic community |
| Morphological Transducer Output | the sporadic nature of press **cover age** of the court often **relegate s it s re port er s** to the **fringe s** of the **journal istic commune ity** |

Table 2: Illustrative example of the same sentence from COCA tokenized orthographically, morphologically, and using BPE.

# B Examples of Surprisal Differences for Morphemes and BPE tokens

Figures 4 and 5 provide two illustrations of surprisal differences between subword segmentations. Note the major difference in the surprisal of *bulb* when summed over BPE tokens when compared to morphological units bulb s. In the sentence in Figure 4, the GPT2 tokenizer split *tulips* into tul ips and did not split *bulbs*. It is reasonable for a human comprehender to expect the word *bulb* immediately after *tulip* since they would cooccur frequently in text, but it is less predictable after lip. This is reflected in the higher surprisal under the BPE-based model.

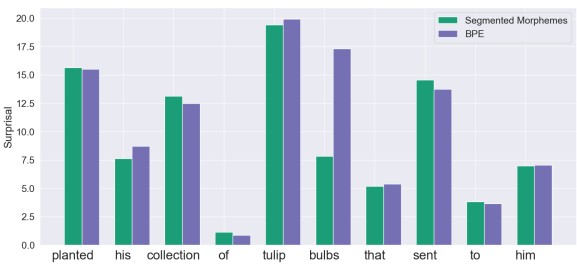

Figure 4: Differences in surprisal of a plural *(bulbs)* under $n$-gram models using morphological and BPE-based tokenization, from a sentence in the Natural Stories corpus.

To take a more morphologically complex example, this difference also occurs for *carefully* in

| Corpus | Tokens Per Word | Percent of Corpus | Percent of Corpus Excluding Stopwords |
|---|---|---|---|
| Dundee | 1 | 94.4 | 88.5 |
| | 2 | 4.19 | 8.68 |
| | 3 | 1.22 | 2.53 |
| | 4 | 0.104 | 0.217 |
| | 5 | 0.005 | 0.011 |
| Natural Stories | 1 | 95.2 | 89.9 |
| | 2 | 3.85 | 8.01 |
| | 3 | 0.971 | 2.02 |
| | 4 | 0.016 | 0.03 |

Table 3: Words in the psycholinguistic corpora split into different numbers of tokens by the BPE tokenizer.

| Corpus | Tokens Per Word | Percent of Corpus | Percent of Corpus Excluding Stopwords |
|---|---|---|---|
| Dundee | 1 | 75.7 | 55 |
| | 2 | 21 | 38.3 |
| | 3 | 3 | 6.18 |
| | 4 | 0.218 | 0.451 |
| | 5 | 0.011 | 0.022 |
| Natural Stories | 1 | 76.9 | 58.3 |
| | 2 | 20.9 | 37.1 |
| | 3 | 2.05 | 4.27 |
| | 4 | 0.125 | 0.26 |

Table 4: Words in the psycholinguistic corpora split into different numbers of tokens by the morphological segmenter.

Figure 5, which is not split up by the BPE tokenizer. The morphological transducer, on the other hand, split *carefully* into care ful ly, producing a much lower surprisal. This suggests that further item-wise comparisons involving words with more morphologically relevant units may be worth investigating.

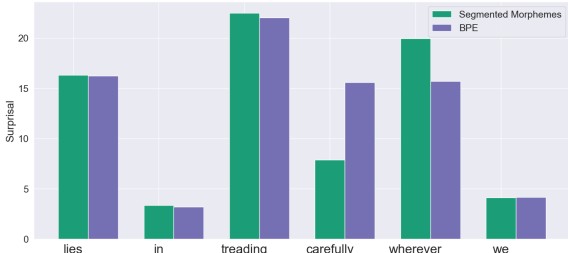

Figure 5: Differences in surprisal of an adverb *(carefully)* under $n$-gram models using morphological and BPE-based tokenization, from a sentence in the Dundee corpus.

## C    Statistical Testing for Predictive Power of Surprisal

### C.1    Effect Sizes

In Table 5 we report effect sizes for the regression models trained on the full psycholinguistic corpora via the widely used Cohen's $f^2$. However, since our work was replicating a subliterature that almost exclusively uses $\Delta LogLik$, we used that measurement in the main body of the paper. We refer readers to Goodkind and Bicknell (2018) for further statistical justification of that choice.

| Tokenization Method | $f^2$ for Dundee | $f^2$ for Natural Stories |
|---|---|---|
| Orthographic | 0.021 | 0.018 |
| BPE | 0.021 | 0.02 |
| Morphological | 0.021 | 0.017 |

Table 5: Effect size comparing surprisal as a feature of regression models relative to controls.

### C.2    Statistical Significance Testing

For the aggregate analysis (Figure 1, we used a Wilcoxon rank-sum test to compute significance. We find no statistically significant difference between the $\Delta LogLik$ estimated for the folds for the two subword tokenization methods relative to predictions over orthographic words.

| Tokenization | Corpus | $W$ | $p$ |
|---|---|---|---|
| BPE | Dundee | 43 | 0.63 |
| Morphological | Dundee | 50 | 1 |
| BPE | Natural Stories | 41 | 0.53 |
| Morphological | Natural Stories | 53 | 0.85 |

Table 6: Wilcoxon rank-sum test comparing distributions of $\Delta LogLik$ from cross-validation results of morphological and BPE versus orthographic surprisal.