# OpenReview forum: "Words, Subwords, and Morphemes: What Really Matters in the Surprisal-Reading Time Relationship?"
_EMNLP/2023/Conference — EMNLP 2023 Findings_

### Official Review · Reviewer_MAqK · 2023-07-25

**Soundness:** 4

**Excitement:**

4: Strong: This paper deepens the understanding of some phenomenon or lowers the barriers to an existing research direction.

**Paper Topic And Main Contributions:**

This paper sets out to investigate an assumption that comes with using LLMs over psycholinguistic data. It specifically examines whether LLM-based predictions based on subword tokenization, rather than decomposition of words into morphemes matters.  The authors compare
surprisal estimates using orthographic, morphological, and BPE tokenization against reading time data. Various publicly available resources are used, such as the COCA (for training a 5-gram model), the Dundee corpus and the Natural Stories corpus (for eye movement data).

**Reasons To Accept:**

This paper have several merits that would benefit both cognitive scientists and the NLP community. On the cognitive science/psycholinguistic side, the authors are testing whether morphological units are a significantly better fit for human measurements. If this turns out to be the case, this would be a strong result, making cognitive researchers/psycholinguistics using pre-trained LLMs aware of the real challenges.

On the NLP side, the authors have used state-of-the art language models (e.g. models using BPE, and an n-gram model based on the output of a morphological transducer), presented their method very clearly, and compared the results with eye movement data.

One result of the work is that surprisal improves predictions of reading time (relative to the control features), which is consistent with previous research. This is shown under each tokenization scheme (evaluated using 10-fold cross-validation for each corpus). The main finding over both corpora is that the predictive power of surprisal in the models using BPE and the morphological transducer is not statistically significant different from the model using orthographic words.

All in all, the authors have shown, quite convincingly, that there's no difference in the predictive power of surprisal associated with tokenization. This is a good result particularly for cognitive scientists.

As the authors themselves report, similar tests should be run over typologically different languages that are morphologically richer than English. Given the clarity of the method, the work can be replicated by future researchers over new languages, given that there's publicly available human data for those languages, such as eye movement data.

**Reasons To Reject:**

I don't have any reasons to reject this paper.

**Reproducibility:**

5: Could easily reproduce the results.

**Reviewer Confidence:**

4: Quite sure. I tried to check the important points carefully. It's unlikely, though conceivable, that I missed something that should affect my ratings.

---

> ### Author Rebuttal · Authors · 2023-08-28
>
> We very much appreciate your positive comments, particularly on the relevance of this work to both cognitive scientists and the NLP community. That said, we would refer you to our responses to the other reviewers. The additional, finer-grained analysis we have done, a closer look at what’s happening at the level of segmented versus non-segmented words, turns out not to be an overall vindication of LLM-style tokenization for cognitive researchers.
>
> In our view, however, this additional result does not undermine the main message of the paper, which is that LLM-style tokenization, with its differences from morphological segmentation, should not simply be accepted blindly as people study the potential value of LLMs in cognitive research and their relationship to human processing. Indeed, we believe the results are now rather more important and interesting: even if there are more superficial or aggregated correspondences, one cannot simply “check the box” and we believe that compared to our original result, these more nuanced results (obtained in response to reviewer feedback) will encourage more rather than than less of the important discussion that should be taking place about LLM-related work and its relationship to cognitive modeling.
>
> Finally, we agree wholeheartedly that extending this work to languages with different morphological systems is an important direction for future research. We will make sure to reinforce this point in the final version if accepted.

---

### Official Review · Reviewer_jJ46 · 2023-08-03

**Typos Grammar Style And Presentation Improvements:** N/A
**Soundness:** 4

**Excitement:**

3: Ambivalent: It has merits (e.g., it reports state-of-the-art results, the idea is nice), but there are key weaknesses (e.g., it describes incremental work), and it can significantly benefit from another round of revision. However, I won't object to accepting it if my co-reviewers champion it.

**Missing References:**

N/A

**Paper Topic And Main Contributions:**

This paper aims to address the question of whether it is a problem for the use of LLMs in surprisal theory that LLMs are trained on subword token prediction while humans do morphological segmentation. The authors compare the predictive power of surprisal in 5-gram models trained on segmented morphemes and on subword tokens and show that there is no significant overall difference in the predictive power of the two models, though they give some examples in the appendix in which the morphological segmentation approach fares better. Though the authors conclude that the use of subword tokenization doesn't have catastrophic implications for the uses of LLMs in psycholinguistic research, they argue that morphological segmentation is still important and should be further explored.

**Questions For The Authors:**

If this paper is accepted, I'd like to see the authors address the questions I outlined above.

**Reasons To Accept:**

The paper is, to my knowledge, the first to address the apparent differences between segmentation between humans and LLMs. This is an important question to be asked, and I hope that if the paper is accepted, it will stimulate further research in this direction.

**Reasons To Reject:**

Given the track the work was submitted to, I'd have liked to see the authors take a firmer stance on the cognitive implications of their findings. Certainly, just because subword tokenization approaches predict similarly well to morphological segmentation approaches doesn't mean that they are equally cognitively plausible. It's unclear to me where this leads us: if we know that LLMs are doing processing differently than humans, why are we using them to predict reading times at all? Do the authors think that incorporating morphological segmentation would solve these problems, and if so how? Would an LLM making use of morphological segmentation be a plausible cognitive model? I realize that these are challenging questions to answer but the nature of a paper exploring such questions is that it is significantly weakened by the authors failing entirely to address any of them.

I'd also like to see the authors address the overall predictive power of surprisal that they report, which seems extremely low across the models -- low enough that I wonder if there may have been methodological issues at play. If the authors are confident in these numbers, then it seems that there are implications more broadly for surprisal theory, and I'd like to see these discussed. Are we really confident in a theory with a predictive power of 0.01?

I think the concerns I raised could be addressed in the additional page with extra discussion, but my main hesitations about this paper at the moment are (1) there may be methodological issues leading to the low predictive powers reported and (2) it lacks discussion of theoretical implications of the findings.

**Reproducibility:**

4: Could mostly reproduce the results, but there may be some variation because of sample variance or minor variations in their interpretation of the protocol or method.

**Reviewer Confidence:**

4: Quite sure. I tried to check the important points carefully. It's unlikely, though conceivable, that I missed something that should affect my ratings.

---

> ### Author Rebuttal · Authors · 2023-08-28
>
> We thank the reviewer for their detailed comments and concerns about the work, and we appreciate your noting this is the first work to address the question of human/LLM segmentation differences. We agree the question is important and share your hope that it will stimulate further research in this direction. If accepted, we will certainly address both major questions in the final version.
>
> Regarding low predictive power for surprisal, the values we report are consistent with those reported by Wilcox et al (2020), who evaluated 5-gram models on the same corpora using the same metric (also around 0.01), providing some degree of confidence that we have not introduced unexpected methodological glitches. All that said, the reviewer is right to question how much of language processing is accounted for by predictive processing, operationalized as surprisal; this is an active direction of research we are currently pursuing, along with many others in the field. We will include relevant discussion in the final version if accepted.
>
> Regarding broader theoretical discussion, there is good evidence that prediction is one element in a theory of human language comprehension (cf. strong correspondence of human neural recordings with LLM outputs in Goldstein et al 2022), and LLMs are useful estimators of the predictability of upcoming linguistic information; however, to be clear, we do not take LLMs themselves to be cognitive models, and we would not take that position even with extremely strong correlations between surprisal and indices of human effort. Surprisal, operationalized using an LLM or any other probability estimates, is at best part of an explanation at Marr’s “computational” level, with no direct connection to algorithms/mechanisms or representations, things that we take to be essential elements of a cognitive model in the sense under discussion here.  In our view, this sub-field has come to rely too much on LLM-based surprisal estimates, especially over words, which do not always correspond to the units humans are predicting (Krauska & Lau, 2023; Brodbeck et al, 2022). This issue is part of what motivated this study to begin with.  Evaluating more explicit models (including algorithms and representations) with behavioral and neural data, as well as developing newer models, are critical directions for future research that we are currently exploring, along with others in the field. One such direction we are pursuing, for example, involves the model of Futrell, R., Gibson, E., & Levy, R. P., 2020,. Lossy‐context surprisal: An information‐theoretic model of memory effects in sentence processing. Cognitive science, 44(3), e12814, within which the incorporation of memory effects presents an opportunity for including explicit psychological models of memory and/or attention.
>
> Even with regard to the present paper, our finer-grained analysis (see response to reviewer aGqM) demonstrates that, despite the results indicating the BPE-based surprisal is solidly correlated with reading times in the aggregate, it differs in an potentially important way from morphological segmentation when we look more closely at non-segmented versus segmented words. We view the results from this finer-grained analysis, in response to our reviews, as a step in the direction of the kind of cognitive model in which we are interested.  Those results also suggest going more deeply into the role of morphological segmentation on an item-by-item basis; for example, training an LLM on morphological units and further following Wilcox et al. (2020) in running on test suites that evaluate (morpho)syntactic generalization.
>
> Again, we very much appreciate your careful attention to the paper and your recommendations with regard to the theoretical and cognitive implications of the work. The discussion above offers a sense of how we are thinking about these issues and we agree with you that it should be possible to incorporate relevant discussion given the additional space in the final version.

---

### Official Review · Reviewer_aGqM · 2023-08-11

**Soundness:** 3

**Excitement:**

2: Mediocre: This paper makes marginal contributions (vs non-contemporaneous work), so I would rather not see it in the conference.

**Paper Topic And Main Contributions:**

This paper tests whether the tokenization process in current NLP practices underlyingly affects the probabilities of words to the extent that computational linguists might extract distorted insights from the evaluation of large language models. This research question was operationalized by looking at surprisals composed of different tokenization methods and testing if the surprisals can help improve the predictability of word reading times, as existing research showed. The findings suggest that the three ways of tokenization all contributed to the prediction. The research provided a first validation and indicated that it is safe to proceed with the status quo tokenizations for LLM-based psycholinguistic studies.

**Questions For The Authors:**

Question 1:
What is the purpose of reporting the per-token log likelihood on a held-out test set using 10-fold cross validation? The contribution/motivation could be stated more clearly.

**Reasons To Accept:**

The strength of the paper is that it tapped into the inner workings of language models and tried to find the connection between its inner working and the validity to evaluate language models as psycholinguistic subjects. It is always an interesting question to ask how tokenization affects training, and testing, and how that affects metrics that parallel human sentence processing. I believe that looking at surprisals and the relation to reading time is one of the operational validation strategies. It would be exciting to see more validation works come around.

**Reasons To Reject:**

1. The judgment criterion could be improved. It seems that this paper relied on whether the surprisal factor significantly helped with the prediction of the reading time in regression models as a criterion to judge the validity of the surprisal measure. More fine-grained or direct criteria could be checked. For example, was the effect size of the surprisal factor across the various surprisals similar? If not, what conclusions could be generated?

2. It is interesting that the tokenization/subword information is studied, but I would appreciate seeing more discussion about why you chose the three ways of tokenization in addition to other existing ones. It feels like the choice was random but I am sure there was a valid reason.

**Reproducibility:**

4: Could mostly reproduce the results, but there may be some variation because of sample variance or minor variations in their interpretation of the protocol or method.

**Reviewer Confidence:**

4: Quite sure. I tried to check the important points carefully. It's unlikely, though conceivable, that I missed something that should affect my ratings.

**Typos Grammar Style And Presentation Improvements:**

The "Method" part could be improved with more clarity. One way that helped with my understanding could be separating into clearer sections, eg. Language Models, Data, Comparison, etc. and making sure that each section only discussed the relevant part. Currently, the "Language Models" part not only missed the model selection (which was stated in the preceding text) but also included data description. It could be improved further.

---

> ### Author Rebuttal · Authors · 2023-08-28
>
> We would like to thank the reviewer for their interest in this work and their helpful comments and suggestions.
>
> Regarding the judgment criterion, our intent, consistent with short-paper criteria, was a small, focused contribution: since the focus here is on tokenization, we vary only that, and keep everything else, including surprisal as the metric, consistent with the existing methods in the literature. Determining whether incorporating surprisal into a regression model improves its estimates of reading times is standard practice for such analyses (Smith & Levy, 2013, current practices use metrics from Goodkind & Bicknell, 2018 and Wilcox et al, 2020). That said, we completely agree that finer-grained analyses are helpful and motivate future research into the role of morphemes in lexical prediction, as suggested by both you and Reviewer jJ46, as well as being of further interest for cognitive scientists, as Reviewer MAqk notes. As discussed below, we have now done some additional fine-grained analysis as suggested, and although status quo tokenizations for LLM-based psycholinguistic studies look safe in the aggregate based on surprisal, the finer-grained results indicate a more nuanced and interesting story.
>
> As for the specific metrics we used, both our methodology and results were consistent with previous published literature: Wilcox et al (2020) reported comparable values for Delta(LogLik) on both corpora we tested, this was also the case with Goodkind & Bicknell (2018)’s findings for the Dundee corpus. In addition, we can add brief findings on effect sizes. Using Cohen’s f^2, we found comparable effect sizes of the surprisal feature across models and corpora (around 0.02), when we compared the models using surprisal with models fitted only for the control features. We will report these statistics if the paper is accepted, and please also see Reviewer jJ46’s comment on small effect sizes, as well as our responses, for further discussion.
>
> In finer-grained analysis, we investigated the extent to which tokenization was actually taking place. As it turns out, the BPE tokenizer only split 5% of the tokens in the psycholinguistic corpora (11% when ignoring stopwords), compared to ~25% and ~44% respectively for the morphological analyzer. Moreover, surprisal theory would suggest processing effort increases incrementally with the number of units a word is segmented into. However, this appears to be true onlyfor the morphological tokenization: for BPE tokenization there is a sharp jump from low surprisal with unsplit words to essentially equal surprisal for words split into 2, 3, and 4 tokens.  In addition, replicating our experiment in a finer-grained way, separately for non-segmented and segmented words, the predictive power of the BPE-based model is worse for words that get split by the tokenizer, but this is not true for the morpheme-based model. Taking all of this together, the finer-grained results suggest that, notwithstanding our aggregate results, there are, in fact, important differences between BPE and morphological segmentation with regard to surprisal. (These were already hinted at by Appendix B in the submission.) If accepted, we will flesh this out, including numbers and figures as appropriate.
>
> Regarding the choice of tokenization schemes, we used orthographic tokenization as a baseline. We chose this because it is how the stimuli were presented to the human participants. N-gram based studies like Smith & Levy (2013) did not consider how words were split, but since LLM-based estimates are now the state of the art, it is important to determine how credible they are because they operate over tokens, which may or may not necessarily correspond to words. We chose the implementation of BPE tokenization used by GPT2, since Oh & Schuler (2022) and Shain et al (2022) showed that larger models were worse predictors of reading times. These details were mentioned in a footnote, but we will promote them to the main text if accepted. Since morpheme-level information has been shown as a component of lexical prediction in humans (Gwilliams, 2020), we used a state-of-the-art morphological transducer that outperformed conventional approaches and LLM tokenizers in the 2022 SIGMORPHON shared task on morphological segmentation. We will make these details clearer in the final version of the paper.
>
> Regarding why we reported cross-validated measurements of the models’ predictive power, we wanted to replicate the findings of Smith and Levy (2013) and Wilcox et al (2020), but only varying tokenization for the n-gram models. This means that we would also need to use the same evaluation metrics. We reported predictive power per token, instead of the aggregate value as in Goodkind & Bicknell (2018), because we were working with multiple corpora as in Wilcox et al (2020). Reporting the value of a cross-validated regression model is important to ensure that the predictive power measures computed on the complete dataset are not the result of overfitting. We will make these motivations clearer in the final version if accepted.
>
> We appreciate the comments on the presentation of the Methods section. We will prioritize ensuring all the relevant details get discussed in their appropriate subsection, as well as having more appropriate subsection titles, if accepted.

---

### Meta-Review · Area_Chair_zLvZ · 2023-09-12

**Recommendation:** 4

**Metareview:**

The paper in question investigates the impact of tokenization methods on the predictability of word reading times in large language models (LLMs). It compares the use of subword tokenization to morphological segmentation and assesses their effectiveness in psycholinguistic research. They provide evidence that predictions using BPE tokenization do not suffer relative to morphological and orthographic segmentation. While the paper addresses an important question, there are notable concerns regarding methodology, theoretical implications, and the overall quality of the results.

Pros from the reviews:

- Relevant Research Question: The paper addresses a pertinent question about the influence of tokenization methods on the predictability of reading times in LLMs, which is of interest to both cognitive scientists and the NLP community.

- Clear Methodology: One reviewer praises the clarity of the paper's methodology and its presentation of results, which contributes to the ease of replication and future research in this area.

- Important Implications: The study's findings have implications for understanding the cognitive processes underlying LLMs' language understanding and their utility in psycholinguistic research.

Cons from the reviews:

- Methodology and Results Concerns: A reviewer raises concerns about the judgment criteria used in the paper, suggesting that more nuanced criteria could be employed to evaluate the impact of tokenization methods. Another reviewer expresses doubts about the overall predictive power of the surprisal models and calls for more rigorous exploration of theoretical implications. Both reviewers question the cognitive plausibility of LLMs in predicting reading times.

- Lack of Theoretical Discussion: A reviewer criticizes the paper for failing to address the theoretical implications of its findings. The reviewer questions the use of LLMs if they differ significantly from human cognitive processing and suggests that the paper should delve into these theoretical questions.

- Limited Scope: While a reviewer appreciates the clarity and presentation of the paper, they point out that the study focuses on English and suggests extending the research to typologically different languages with richer morphology. This limitation reduces the generalizability of the findings.

I have read the rebuttals and followed the changes after each acknowledgement. The rebuttals helped to clarify many of the reasons to reject the paper. Overall, the conference could benefit from such a short paper contribution, but this short paper contribution could also benefit from some additional content that has been raised in the rebuttal discussions. Initially, various point have been unclear and although the authors could clarify them, the paper could benefit from some additional content. Overall, all reviewers would like to see the paper as a contribution to the field and I can agree with that.

---

### Decision · Program_Chairs · 2023-10-07

**Decision:**

Accept-Findings

**Comment:**

The paper in question investigates the impact of tokenization methods on the predictability of word reading times in large language models (LLMs). It compares the use of subword tokenization to morphological segmentation and assesses their effectiveness in psycholinguistic research. They provide evidence that predictions using BPE tokenization do not suffer relative to morphological and orthographic segmentation. While the paper addresses an important question, there are notable concerns regarding methodology, theoretical implications, and the overall quality of the results.

Pros from the reviews:

- Relevant Research Question: The paper addresses a pertinent question about the influence of tokenization methods on the predictability of reading times in LLMs, which is of interest to both cognitive scientists and the NLP community.

- Clear Methodology: One reviewer praises the clarity of the paper's methodology and its presentation of results, which contributes to the ease of replication and future research in this area.

- Important Implications: The study's findings have implications for understanding the cognitive processes underlying LLMs' language understanding and their utility in psycholinguistic research.

Cons from the reviews:

- Methodology and Results Concerns: A reviewer raises concerns about the judgment criteria used in the paper, suggesting that more nuanced criteria could be employed to evaluate the impact of tokenization methods. Another reviewer expresses doubts about the overall predictive power of the surprisal models and calls for more rigorous exploration of theoretical implications. Both reviewers question the cognitive plausibility of LLMs in predicting reading times.

- Lack of Theoretical Discussion: A reviewer criticizes the paper for failing to address the theoretical implications of its findings. The reviewer questions the use of LLMs if they differ significantly from human cognitive processing and suggests that the paper should delve into these theoretical questions.

- Limited Scope: While a reviewer appreciates the clarity and presentation of the paper, they point out that the study focuses on English and suggests extending the research to typologically different languages with richer morphology. This limitation reduces the generalizability of the findings.

I have read the rebuttals and followed the changes after each acknowledgement. The rebuttals helped to clarify many of the reasons to reject the paper. Overall, the conference could benefit from such a short paper contribution, but this short paper contribution could also benefit from some additional content that has been raised in the rebuttal discussions. Initially, various point have been unclear and although the authors could clarify them, the paper could benefit from some additional content. Overall, all reviewers would like to see the paper as a contribution to the field and I can agree with that.